# Mirvetuximab Soravtansine in solid tumors: A systematic review and meta-analysis

Shamsnur Rehim[1,2☉], Shuang Yuan[1,2☉], Hongjing Wang[1,2]*

1 Department of Obstetrics and Gynaecology, West China Second University Hospital, Sichuan University, Chengdu, China, 2 Key Laboratory of Birth Defects and Related Diseases of Women and Children of the Ministry of Education, West China Second University Hospital, Sichuan University, Chengdu, China

☉ These authors contributed equally to this work.
* whjscdx@163.com

## Abstract

### Background

Mirvetuximab Soravtansine (MIRV) is a promising antibody–drug conjugate (ADC) that targets folate receptor alpha (FRα), which is overexpressed in several types of solid tumors. In November 2022, MIRV was approved in the USA for the treatment of adult patients with FRα-positive, platinum-resistant epithelial ovarian, fallopian tube or primary peritoneal cancer who received 1–3 prior systemic treatment regimens. Therefore, high-quality evidence for its efficacy and safety in different cancers is urgently needed.

### Methods

A systematic search (e.g., PubMed, Embase, Web Of Science, Cochrane Library) was conducted to identify all relevant clinical trials of MIRV alone or in combination with chemo- and/or target-therapies in solid tumors. The primary end-point was median progression-free survival (mPFS). The secondary endpoints were the Objective response rate (ORR) and adverse effects (AEs). A random-effects model was applied.

### Results

The study included nine research studies with a total of 682 patients. The pooled mPFS and pooled ORR were 6.70 months (95% CI 4.54–8.86, I2 = 96.21%) and 36% (95% CI: 28% to 44%, I2 = 76.79%), respectively. Significant differences were observed among intervention regimens and response to platinum. The pooled mPFS of MIRV monotherapy and MIRV+ Bevacizumab (BEV) combined therapy was 4.28 (95% CI 3.90–4.65, I2 = 0.00%) and 7.78 (95% CI 6.62–8.95, I2 = 0.00%), respectively. The pooled ORRs of MIRV monotherapy and MIRV+BEV combined therapy were 25% (95% CI 21%–29%, I2 = 25.20%) and 43% (95% CI 36%–50%, I2 = 0.01%), respectively. The pooled ORRs of the platinum-sensitive, platinum-resistant groups were 59% (95% CI 36%–81%, I2 = 61.88%), 33% (95% CI 25%–40%, I2 = 69.73%), respectively. In addition, we conducted supplementary subgroup analyses to explore the influence of FRα receptor expression levels and the number of prior treatments

**Data Availability Statement:** All relevant data are within the paper and its Supporting Information files.

**Funding:** The author(s) received no specific funding for this work.

**Competing interests:** The authors have declared that no competing interests exist.

on treatment outcomes. The most common adverse effects were blurred vision (45.20%), nausea (40.13%), diarrhea (39.52%), fatigue (33.84%) and keratopathy (31.20%).

## Conclusions

MIRV has significant therapeutic effects in solid tumors, especially when combined with BEV. In platinum-tolerant tumors, the efficacy of MIRV is also considerable. Overall, MIRV is relatively safe in solid tumors, and adverse reactions are relatively rare and mild.

## 1. Introduction

Antibody drug conjugates (ADCs) have been in use for more than 20 years, primarily for treating hematological malignancies. The concept of ADCs as a "magic bullet" was first introduced by Paul Ehrlich [1], who described the selective delivery of a cytotoxic drug to cancer cells. ADCs are designed to bind to cell surface targets and deliver a cytotoxic payload via internalization, breaking down the linker to release the drug [2]. The ideal mechanism relies on cellular receptor internalization to deliver the cytotoxic agent directly to the tumor cells [3]. ADCs have gained ground in treating solid tumors with the FDA approval of trastuzumab emtansine (T-DM1) in February 2013 for metastatic and early-stage breast cancer [4].

Mirvetuximab Soravtansine (MIRV) is also one of the recently U.S. Food and Drug Administration (FDA) -accelerated ADCs that is used to treat solid tumors. The approval was mainly based on the results of the SORAYA study, which evaluated the efficacy and safety of MIRV in tumors expressing high levels of folate receptor alpha (FRα) [5]. MIRV is a type of anti-FRα ADC that consists of a humanized IgG1 antibody that has been conjugated with a DM4 payload using a cleavable linker [6]. This special type of antibody is designed to selectively target cancer cells that express FRα, leading to internalization of the drug by the cells. FRα is a crucial molecule indispensable for DNA replication and repair and RNA synthesis, as it participates in nucleotide synthesis, amino acid metabolism, and phospholipid biosynthesis [7]. Various epithelial tumors, such as ovarian, endometrial, triple-negative breast, and non-small cell lung cancers, have been shown to exhibit aberrant overexpression of FRα [8]. The overexpression of FRα in cancer cells, along with its ability to bind with nonphysiological substrates and influence cell division and migration, makes it a significant target for cancer treatments [9]. DM4 is a synthetic derivative of maytansine, which is a cytotoxic agent found in certain plants, as a payload can effectively permeate cell membranes and interact with tubulin molecules, causing cells to arrest in the G2-M phase and ultimately leading to their death [10]. Additionally, the payload utilizes the bystander effect to kill neighboring cells [11].

Based on the advantages mentioned above, the application prospects of MIRV in solid tumors expressing FRα are very promising. Although it has received accelerated approval from the FDA, there remains an urgent need to evaluate its efficacy, safety, and other aspects [3]. In particular, the efficacy and safety of combining MIRV with chemotherapy or other targeted agents, including immunotherapy, need to be further analysed. To this end, a meta-analysis was conducted to assess the efficiency and safety of this newly approved drug.

## 2. Materials and methods

We followed Preferred Reporting Items for Systematic Reviews and Meta-Analyses (PRISMA) guidelines for the reporting of Systematic review and meta analysis [12]. A prospectively developed study protocol was registered on PROSPERO (CRD42023413646).

## 2.1 Data identification, selection and extraction

This meta-analysis was performed following the Preferred Reporting Items for PRISMA statement [13] and included all studies reporting relevant clinical trial results without any restriction on publication year. The MEDLINE (via PubMed), Embase(via OvidSP), CENTRAL (via Cochrane Library), Web of Science, CNKI, Wanfang, VIP and ClinicalTrials.gov databases were systematically searched with the terms "MIRV" "IMGN853" "antibody–drug conjugate", "ADC", "solid tumor", and "cancer" to identify all relevant clinical trials of MIRV alone or in combination with chemo- and/or target-therapies in solid tumors published from inception to November 31, 2023. The language was restricted to English and Chinese. In addition, the reference lists of the selected articles were manually searched to identify additional studies.

This systematic review and meta-analysis included clinical trials that investigated the use of MIRV alone or in combination with chemo- and/or targeted therapies for the treatment of any solid tumor, regardless of their trial design (single-arm or randomized controlled trial). Excluded studies were abstracts from meetings, reviews, systematic reviews, meta-analyses, cost-effectiveness analyses, editorials, opinions, or case reports. The study characteristics were extracted, including the first author, publication year, region/country, study design, sample size, and intervention type. The patient characteristics recorded comprised the tumor type, FRα expression level, prior treatment history, and whether the intervention was administered alone or combined with chemo-therapies.

The primary endpoint of the study was Hazard Ratio (HR) with a 95% confidence interval for median progression-free survival (mPFS). The secondary endpoint was the relative risk (RR) with a 95% CI for the objective response rate (ORR) and adverse events (AEs) associated with MIRV administration. The Cochrane Risk of Bias tool for Randomized Controlled Trials (ROB2)_ [14] and the Risk of Bias in Non-randomized Studies—of Interventions (ROBINS-I) tool [15] were used to assess the quality and risk of bias of the included studies.

## 2.2 Endpoints

The outcome indicators were selected based on the guidance documents of the US Food and Drug Administration (FDA) [16]. The primary endpoint was HR with 95% CI for mPFS. The secondary endpoint was the relative risk (RR) with 95% CI for the objective response rate (ORR) and safety profile of MIRV administered alone or in combination. The safety profile included all grades of adverse effects, such as blurred vision, dry eye, nausea, vomiting, fatigue, diarrhea, abdominal pain, anemia, neutropenia, thrombocytopenia, decreased appetite, headache, increased Alanine Aminotransferase(ALT)/Aspartate Aminotransferase(AST), and pneumonia. Patient OS was not considered an endpoint of this meta-analysis, as these data were available for only one trial.

## 2.3 Statistical analysis

Statistical analysis of the pooled OR, mPFS and AE results of patients with solid tumors treated with MIRV was performed using STATA/SE version 16.1 (Stata Corp., TX, USA) [17]. The effect size of all pooled results was represented by 95% CI with an upper limit and a lower limit. The Cochrane Q chi-square test and I2 statistic were used to examine the heterogeneity across studies [18]. The fixed-effects model was used for pooled results with low heterogeneity (I2 ≤ 50%); otherwise, the random-effects model was used for analysis. To assess the stability of the pooled results, multiple sensitivity analyses (influence analysis) were performed [19]. Subgroup analysis was conducted based on whether the treatment was administered alone or in combination withchemotherapy, as well as the sensitivity to platinum. Publication bias was

explored through visual inspection of funnel plots, Egger's regression tests and Begg's tests [20, 21]. A p value <0.05 was considered significant.

# 3. Results

## 3.1 Study characteristics

Out of 493 records that were examined, only 9 studies were found to meet the requirements for inclusion, as shown in Fig 1. The meta-analysis that followed used a total of 10 records (two records from same study) that featured 682 patients. The baseline characteristics of the patient populations were generally similar across the studies.

There were a variety of study designs, including 1 randomized controlled trial [22], 7 single-arm studies [23–30] and 1 multiarm study [25], which provided two records for analysis. The patients who participated in these studies primarily had gynecological cancer, and many

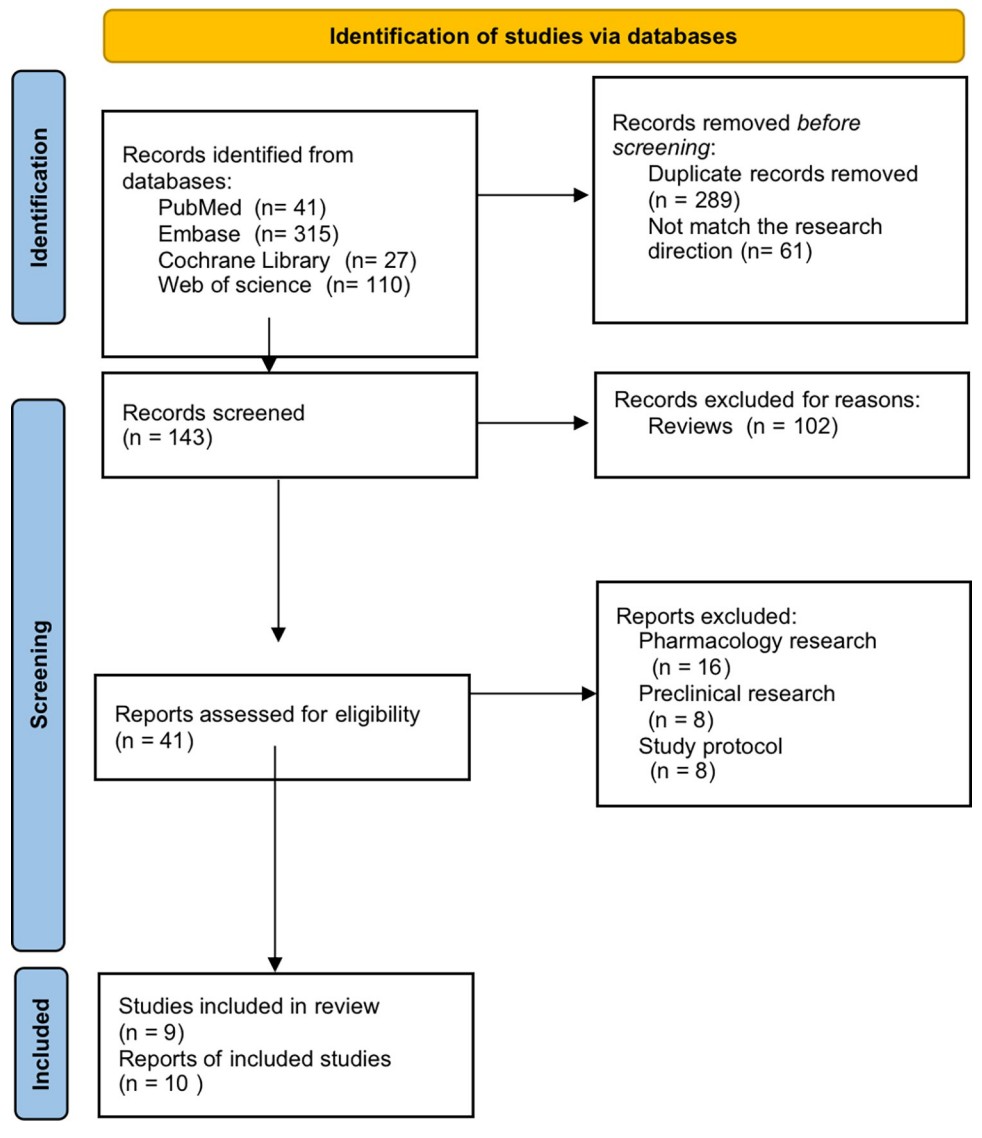

**Fig 1. PRISMA flow chart.**

of them had prior cancer treatment, with some even having confirmed platinum resistance. The intervention administered in these studies was either monotherapy with MIRV or a combination of this drug with other medications that have antitumour effects, such as gemcitabine (GEM), carboplatin (CARBO), and rucaparib. For more information on the study features, please refer to Table 1.

## 3.2 Primary end-point: mPFS

The collective mPFS estimated from the pool of nine records yielded an average span of 4.70 months (95% CI 4.35–5.05). This suggests that patients typically enjoy approximately 6.70 months of disease stability before facing disease progression. Nevertheless, it is important to note that there was significant heterogeneity among the studies, as evidenced by a high I2 value of 96.21%. For a detailed overview of the individual study estimates and their respective confidence intervals, refer to Fig 2. depicting the forest plot.

## 3.3 Secondary end-point: ORR

The objective response rate (ORR) was calculated as the percentage of patients who achieved a complete or partial response to treatment with MIRV. In our meta-analysis, the ORR ranged from 22% to 71%, with a weighted average of 36% (95% CI: 28 to 44). However, there was significant diversity between the studies, with an I squared value of 76.79%. For a detailed overview of the individual study estimates and their respective confidence intervals, refer to Fig 3. depicting the forest plot.

## 3.4 Sensitivity analysis

To test the robustness of our statistical results, we conducted leave-one-out sensitivity analyses. The effect indicators of mPFS was 1.84 (95% CI: 1.55 to 2.13) (S1 Fig), indicating significant

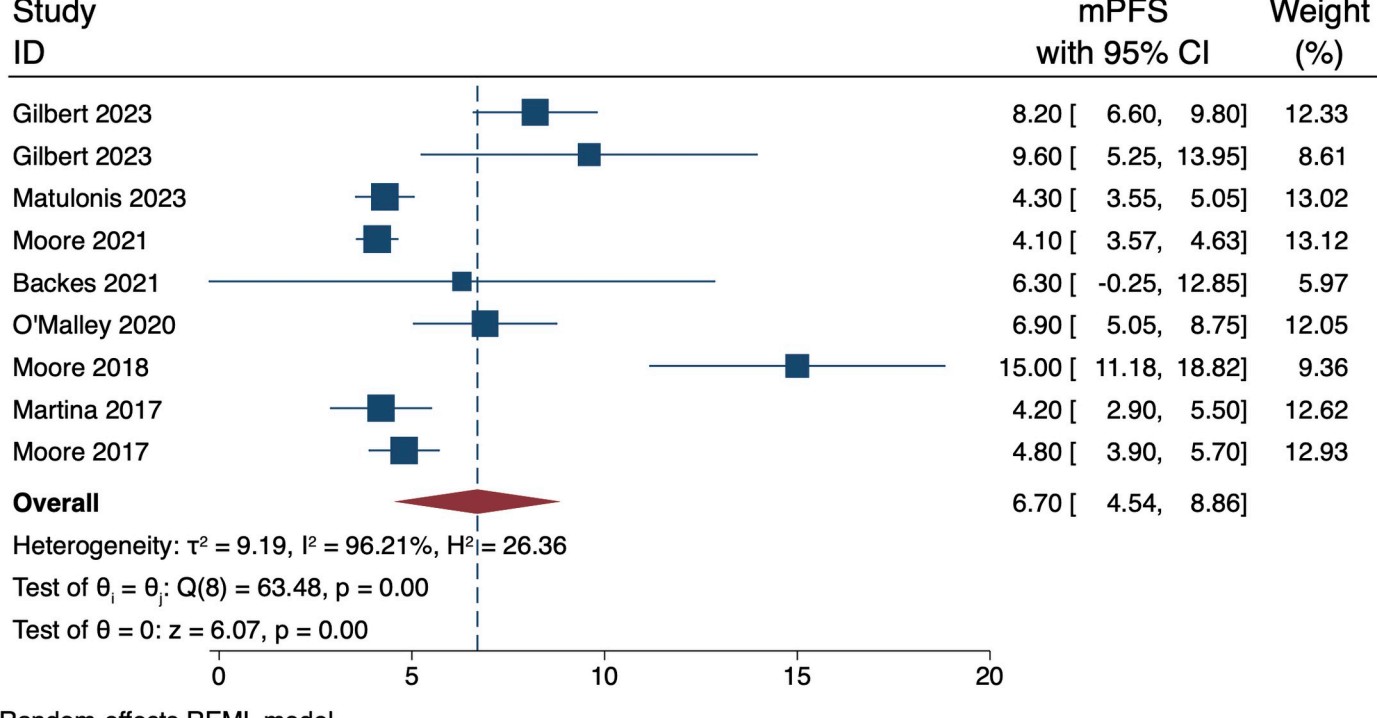

**Fig 2. Forest plot of mPFS.** mPFS: median Progression-Free Survival.

Table 1. Main characteristics of the included studies.

| Study | Country | Study period | Study Type | Follow-up time/months (range) | Cancer type | Sample size | Platinum response status | No. of prior systemic therapies[1], n (%) | Prior compound exposure, n (%) | FRα expression[2], n (%) | Intervention | Age/years, median (range) | End points | Criteria for response | Criteria for AEs |
|---|---|---|---|---|---|---|---|---|---|---|---|---|---|---|---|
| Gilbert 2023 | Belgium, Canada, Spain, United States | 2015–2021 | Multi-arm, Open Label, Phase 1b/2 | 8.7 (0.8–25.3) | Ovarian, Fallopian Tube, or Primary Peritoneal Cancer | 94 | Platinum-resistan | <3—— 45 (48) ≥3—— 49 (52) | Platinum 41 (100) Taxane 91 (97) Bevacizumab 55 (59) PARP inhibitor 25 (27) | High 44 (47) Medium 39 (42) Low 11 (12) | MIRV + BEV | 62 (39–81) | ORR, PFS, DOR, AEs | RECIST | CTCAE |
| | | | | | | 31 | platinum-sensitive | <3—— 23 (74) ≥3——8 (26) | Platinum 41 (100) Taxane 31 (100) Bevacizumab 10 (32) PARP inhibitor 8 (26) | High 18 (58) Medium 12 (39) Low 1 (3) | MIRV + BEV | 59 (44–83) | | | |
| Matulonis 2023 | United States | 2020–2022 | Single-arm, Open Label, Phase II | 13.4 | Ovarian Cancer | 105 | Platinum-Resistant | <3—— 51 (50.5) ≥3——54 (49.5) | Platinum 106 (100) Taxanes 105 (99) Bevacizumab 106 (100) PARP inhibitor 51 (48) Liposomal doxorubicin 75 (71) | High 38 (36) | MIRV | 62 | ORR, PFS, DOR, AEs | RECIST | CTCAE |
| Moore 2021 | 12 countries | 2015–2020 | RCT, phase III | 12.5 (0.03 to 22.0) | Ovarian, Fallopian Tube, or Primary Peritoneal Cancer | 243 | Platinum-resistant | <3—— 159 (65.4) ≥3——86 (34.6) | Platinum 243 (100) Paclitaxel 238 (100) Bevacizumab 121 (48.8) PARP inhibitor 44 (17.7) | High 147 (59.3) Medium 101 (40.7) | MIRV | 64 (34–89) | OS, ORR, PFS, DOR, AEs | RECIST | CTCAE |
| Mihaela 2021 | United States | 2017–2020 | Single-arm, Open Label, Phase I | —— | Epithelial ovarian cancer | 30 | Platinum resistant | ≥3—— 30 (100) | —— | High 15 (30) Medium 10 (33) Low 5 (17) | MIRV+Gem | —— | ORR | RECIST | CTCAE |
| Backes 2021 | United States | 2018–2021 | Single-arm, Open Label, Phase I | —— | Ovarian cancer Endometrial cancer | 21 | —— | ≥3—— 21 (100) | Platinum 21 (100) Paclitaxel 21 (100) PARP inhibitor 21 (100) | —— | MIRV +Rucaparib | 64.5 | ORR, PFS, AEs | RECIST | CTCAE |

(Continued)

**Table 1.** (Continued)

| Study | Country | Study period | Study Type | Follow-up time/ months (range) | Cancer type | Sample size | Platinum response status | No. of prior systemic therapies[1], n (%) | Prior compound exposure, n (%) | FRα expression[2], n (%) | Intervention | Age / years, median (range) | End points | Criteria for response | Criteria for AEs |
|---|---|---|---|---|---|---|---|---|---|---|---|---|---|---|---|
| O'Malley 2020 | Belgium, Canada, Spain, United States | 2015–2019 | Single-arm, Open Label, Phase 1b/2 | 14.8 (0.76–25.3) | Ovarian cancer | 66 | Platinum-resistant | <3 — 27 (41) ≥3 — 39 (49) | Platinum 66 (100) Taxanes 65 (99) Bevacizumab 41 (62) PARP inhibitor 20 (30) | High 28 (42) Medium 24 (36) Low 13 (20) | MIRV + BEV | 63 (39–81) | ORR, PFS, DOR, AEs | RECIST | CTCAE |
| Moore 2018 | Belgium, Canada, Spain, United States | 2015–2018 | Single-arm, Open Label, Phase 1b/2 | 15.9 | Relapsed Ovarian cancer | 18 | platinum-sensitive | <3 — 9 (50) ≥3 — 9 (50) | Platinum 18 (100) Taxane 18 (100) Bevacizumab 5 (28) PARP inhibitor 9 (50) | High 7 (39) Medium 4 (22) Low 7 (39) | MIRV + CARBO | 66 (47–82) | ORR, PFS, DOR, AEs | RECIST | CTCAE |
| Moore 2017 | Canada, United States | 2014–2016 | Single-arm, Open Label, Phase I | 5.5 | Ovarian, Fallopian Tube, or Primary Peritoneal Cancer | 46 | Platinum-Resistant | ≤3 — 23 (50) ≥3 — 23 (50) | Platinum 21 (100) Paclitaxel 21 (100) | High 23 (50) Medium 14 (30.4) Low 9 (19.6) | MIRV | 62.5 | ORR, PFS, DOR, AEs | RECIST | CTCAE |
| Martina 2017 | Canada, United States | 2012–2017 | Single-arm, Open Label, Phase I | — | Relapsed Ovarian, Fallopian Tube cancer | 27 | platinum-sensitive 7 (26) Platinum-Resistant 20 (74) | ≤3 — 10 (37.1) ≥3 — 17 (62.9) | Platinum 27 (100) Taxane 27 (100) Bevacizumab 23 (85) PARP inhibitor 9 (33) | — | MIRV | 62 (38–76) | ORR, PFS, AEs | RECIST | CTCAE |

ORR: Objective Response Rate; PFS: Progression-Free Survival; DOR: Duration of Response; AEs: Adverse Events; RECIST: Response Evaluation Criteria in Solid Tumors; CTCAE: Common Terminology Criteria for Adverse Events

[1] The three numbers provided represent the number of prior systemic chemotherapies the patients received, the number of patients who received that number of treatments, and the percentage of these patients within the total study population, respectively.

[2] Low, ≤50%,; Medium, 50%-74%; High ≥ 75% of tumor cells with any FRa membrane staining visible at 10 microscope objective

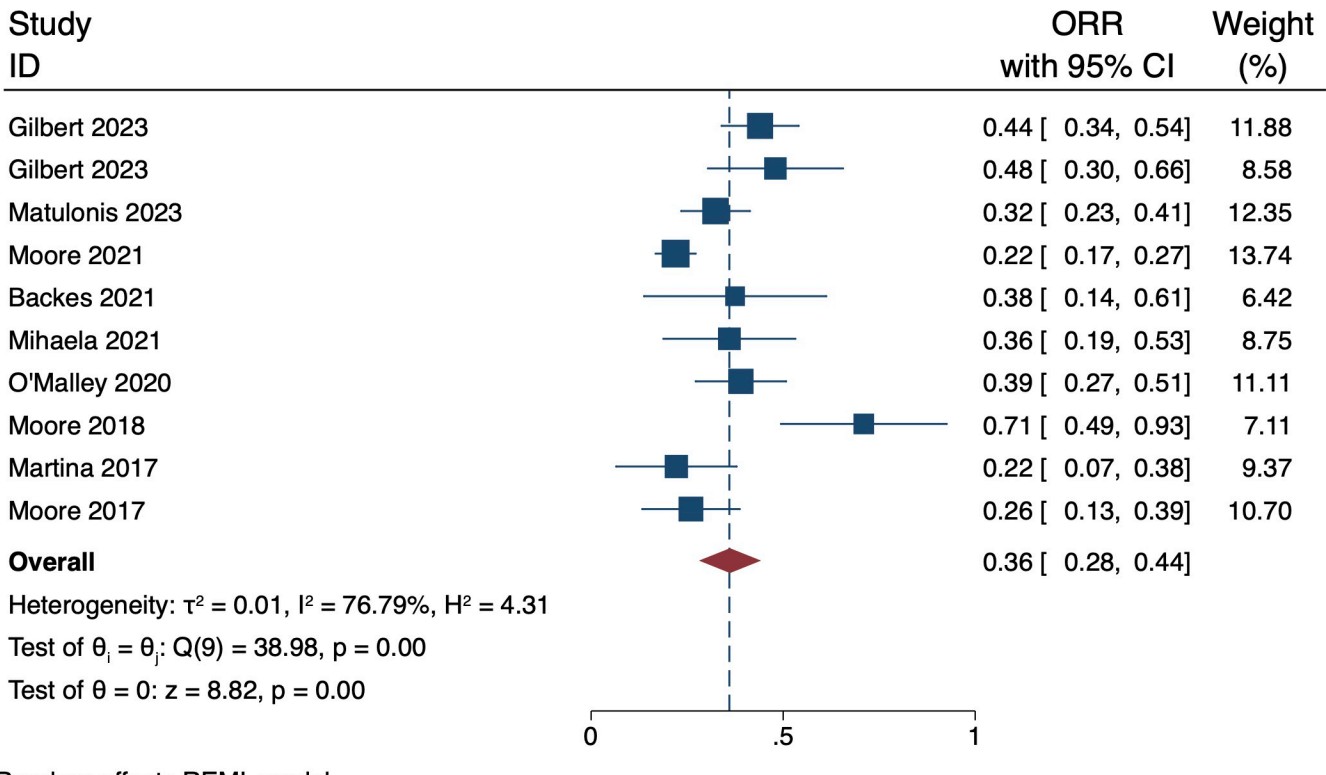

**Fig 3. Forest plot of ORR.** ORR: Objective Response Rate.

heterogeneity. According to sensitivity analysis, Moore 2018 was the study with the greatest impact on the results. A pooled analysis was carried out after excluding Moore 2018, which resulted in a pooled mPFS and ORR of 4.61 months (95% CL: 4.25–4.97) and 30% (95% CL: 26% to 33%), respectively, still showing relatively significant heterogeneity between studies (S2 Fig)

### 3.5 Secondary end-point: Safety

In assessing the safety of MIRV, it is crucial to consider adverse reactions as an outcome indicator. To evaluate such reactions, we classified them into two levels–any and 3/4 –for statistical analysis and found that adverse reactions were mostly consistent across studies. All studies utilized the Common Terminology Criteria for Adverse Events (CTCAE) grading system to evaluate adverse reactions. The percentage of patients experiencing a particular adverse event was used to evaluate the adverse events in all the studies. The analysis showed that the most common adverse effectswere blurred vision (45.20%), nausea (40.13%), diarrhea (39.52%), fatigue (33.84%) and keratopathy (31.20%). However, few grade 3 and 4 adverse effects have occurred. Thrombocytopenia (4.76%) and increased levels of ALT in the blood (3.09%) were the most frequently reported grade 3 and 4 adverse reactions. Table 2 shows more details about the safety analysis.

### 3.6 Sub-group analysis

To further analyse the sources of heterogeneity in the study, we conducted subgroup analysis based on the drug use patterns and sensitivity to platinum included in the study. Our subgroup

**Table 2. Pooled results of common AEs of any grade and ≥grade 3.**

| AEs | Any grade | | | 3/4Grades | | |
|---|---|---|---|---|---|---|
| | ES, % (95% CI) | I², % | N* | ES, % (95% CI) | I², % | N* |
| Vision blurred | 45.20(39.32–51.09) | 45.18 | 9 | 2.17 (0.96–3.38) | 44.12 | 5 |
| Nausea | 40.13 (32.27–47.99) | 70.57 | 9 | 1.28 (0.29–2.28) | 0.06 | 5 |
| Diarrhea | 39.52 (28.86–50.18) | 84.98 | 8 | 1.89 (0.84–2.95) | 0.02 | 8 |
| Fatigue | 33.84 (26.90–40.77) | 64.25 | 9 | 1.58 (0.62–2.54) | 0.02 | 8 |
| Keratopathy | 31.2 (27.45–34.94) | 0.03 | 7 | 1.66 (0.34–2.99) | 86.50 | 2 |
| Neuropathy | 27.33 (19.46–35.19) | 76.61 | 7 | 1.81 (0.48–3.14) | 0.00 | 3 |
| AST increased | 23.58 (18.29–28.87) | 36.60 | 7 | 1.76 (0.55–2.97) | 26.52 | 4 |
| Dry eye | 23.30(18.70–27.90) | 37.45 | 7 | 1.47 (0.43–2.52) | 0.00 | 4 |
| Thrombocytopenia | 23.17 (12.01–34.34) | 82.59 | 5 | 4.76 (1.82–7.70) | 0.00 | 4 |
| ALT increased | 22.19(17.29–27.09) | 0.01 | 6 | 3.09 (0.73–5.44) | 0.00 | 3 |
| Headache | 20.29(12.52–28.06) | 54.79 | 5 | - | - | - |
| Vomiting | 19.54(13.79–25.28) | 62.74 | 7 | 1.24 (0.22–2.27) | 0.00 | 4 |
| Decreased appetite | 18.71(12.58–24.85) | 66.80 | 6 | 0.85 (-0.11–1.82) | 0.00 | 2 |
| Anemia | 12.96(9.41–16.51) | 81.08 | 4 | 1.02 (-0.06–2.09) | 39.57 | 4 |

AEs: Adverse Events

* Number of studies considered for pooling the safety events

analysis showed that the combination of MIRV and BEV had a markedly more favorable effect than either agent alone, with an mPFS of 7.78 months compared to 4.28 months for single use (Fig 4). The ORR was also higher in patients who received combination therapy with MIRV and other anticancer drugs (43%) than in those who received MIRV alone (25%) (Fig 5). In addition, according to our subgroup analysis, the ORR of patients with platinum sensitivity to previous chemotherapy (59%) was significantly higher than that of patients with platinum resistance (33%) (Fig 5), and the corresponding pooled mPFS was 12.65 vs 4.60 months (Fig 4). Furthermore, we conducted supplementary subgroup analyses to explore the influence of FRα receptor expression levels and the number of prior treatments on treatment outcomes. Our analysis revealed that patients with high FRα receptor expression (defined as expression levels greater than or equal to 50%) had a pooled overall response rate (ORR) of 47% [27%, 66%] (Fig 5), while those with low expression (defined as expression levels less than 50%) exhibited a pooled ORR of 29% [9%, 49%] (Fig 5). These findings suggest a potential link between FRα receptor expression and treatment response. Moreover, when considering the number of prior treatments, patients who had received 1–2 lines of prior therapy demonstrated a pooled ORR of 43% [23%, 63%](Fig 5), whereas those who had undergone ≥3 lines of prior therapy had a pooled ORR of 34% [24%, 44%] (Fig 5). These results offer valuable insights into the varying treatment responses based on FRα expression levels and the number of prior therapies.

Additionally, due to the limited number of studies reporting data on the relative aspects, it may be necessary to obtain additional relevant randomized controlled trial results to validate these findings. It is important to note that the lack of studies providing mPFS (median progression-free survival) data for these subgroups resulted in the meta-analysis being solely based on ORR values. Therefore, further research is warranted to comprehensively understand the impact of FRα receptor expression levels and the number of prior treatments on treatment outcomes in cancer patients.

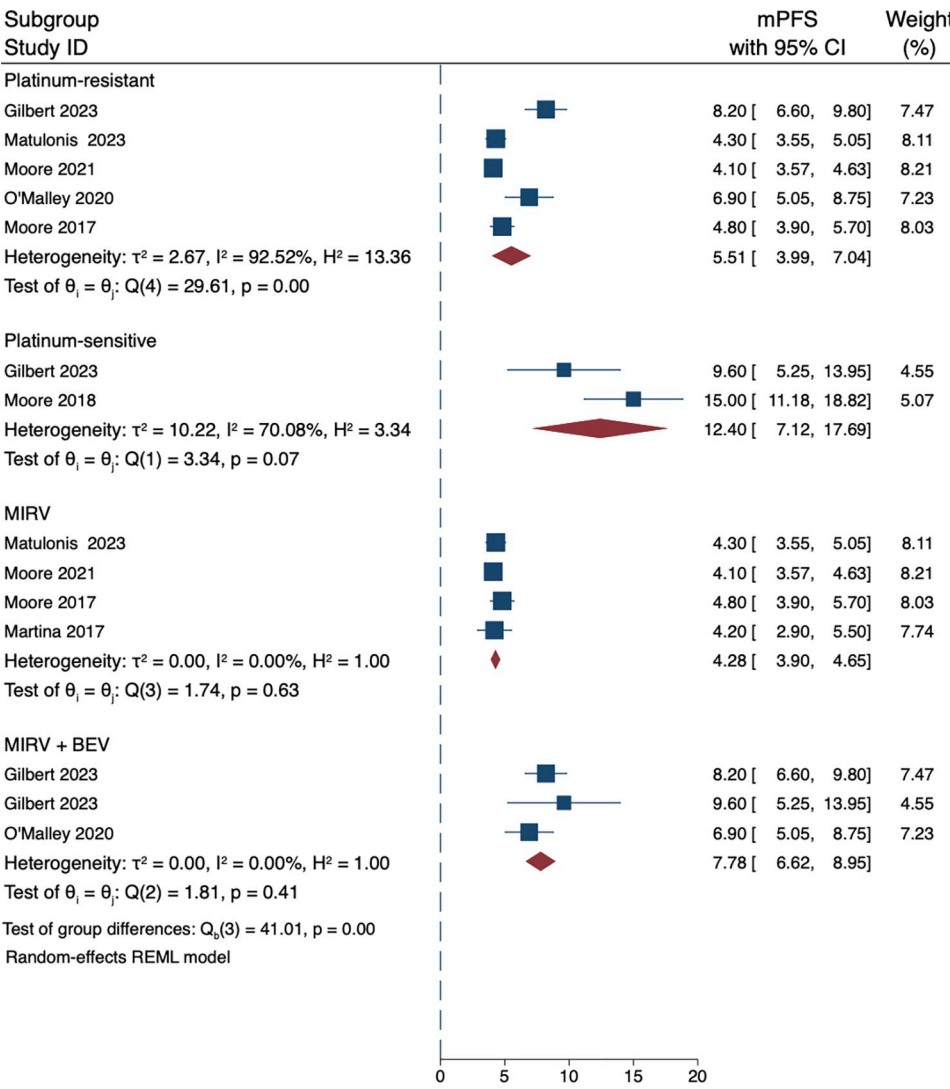

**Fig 4. Pooled mPFS by groups.** mPFS: median Progression-Free Survival.

### 3.7 Risk-of-bias analysis and publication bias

The only randomized controlled study included in our analysis had a low risk of bias, as assessed by the ROB2 tool. To evaluate the bias risk for the remaining eight single-arm clinical studies, we used the ROBINS-I tool, which showed a high risk of bias [Table 3]. A major contributing factor to risk was the presence of confounding factors.

## 4. Discussion

To the best of our knowledge, our meta-analysis is the first article evaluating the efficacy and safety of MIRV in solid tumors. Since the drug itself has been used in clinical settings for a relatively short time and has been approved by the FDA through an accelerated process [31], most of the clinical trials we collected were single-arm trials [22–30], which may have slightly lower quality evidence. Nevertheless, our analysis still yielded relatively encouraging results. We primarily collected and analysed data on ORR, mPFS, safety, etc., after patients received MIRV treatment, regardless of their tumor type, stage, previous treatments, or therapy regimen. The

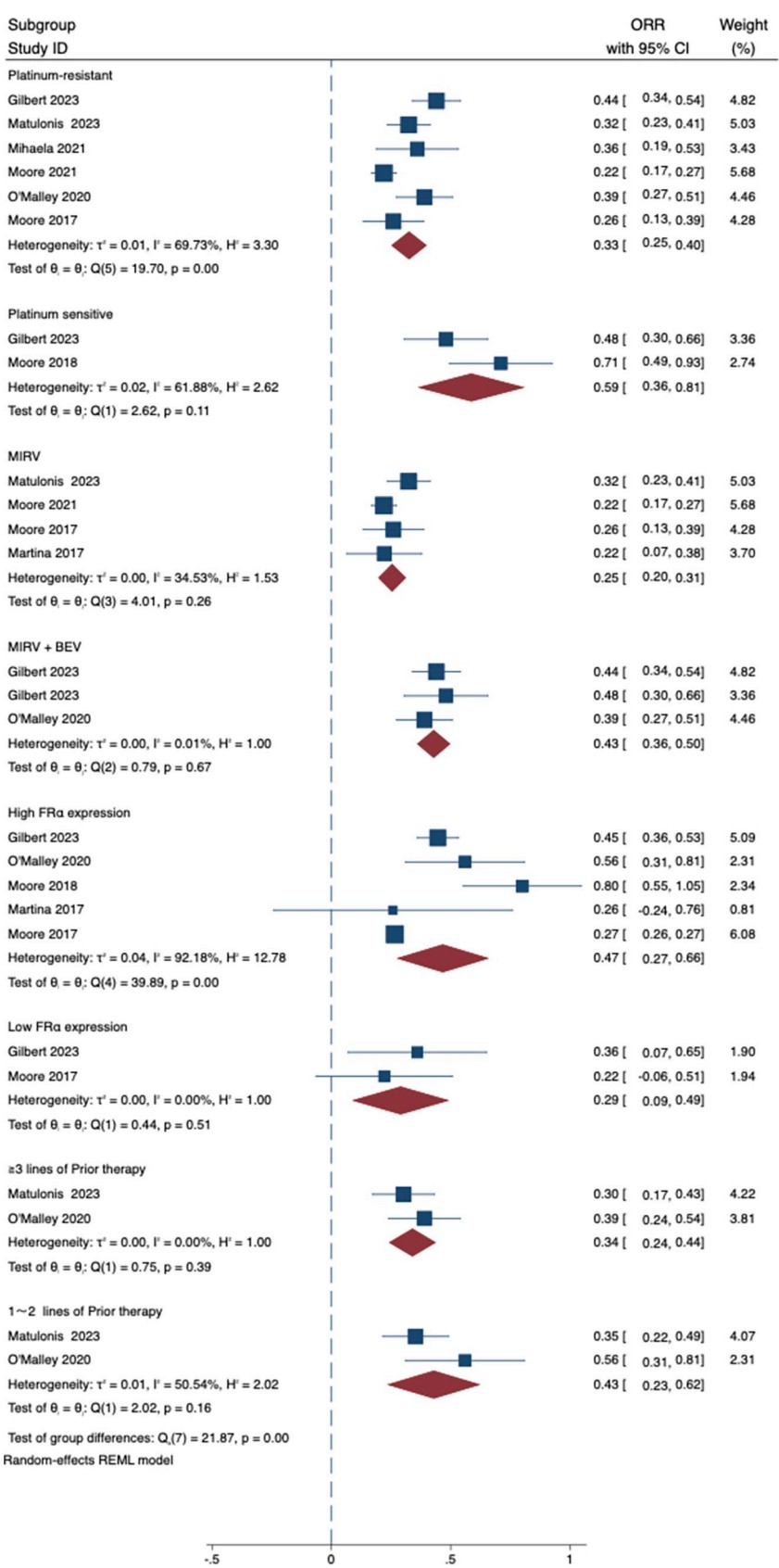

**Fig 5. Pooled ORR by groups.** ORR: Objective Response Rate.

**Table 3. Quality assessment of included studies.**

| A. ROBINS-I tool for included single arm trails | | | | | | | |
|---|---|---|---|---|---|---|---|
| Study ID | Domain 1 | Domain 2 | Domain 3 | Domain 4 | Domain 5 | Domain 6 | Domain 7 | Overall bias |
| Matulonis 2023 | Serious | low | low | low | Moderate | Moderate | low | Serious |
| Gilbert 2023 | Serious | low | low | low | low | Moderate | low | Serious |
| Mihaela 2021 | Serious | low | low | low | low | Moderate | Moderate | Serious |
| Backes 2021 | Serious | Moderate | low | Moderate | Serious | Serious | low | Serious |
| O'Malley 2020 | Serious | low | low | low | low | Moderate | Moderate | Serious |
| Moore 2018 | Serious | low | low | low | low | Moderate | low | Serious |
| Moore 2017 | Serious | low | low | low | low | Moderate | Moderate | Serious |
| Martina 2017 | Serious | low | low | low | low | Moderate | low | Serious |

| ROB 2 tool for included RCT | | | | | |
|---|---|---|---|---|---|
| Study ID | Randomization process | Deviations from intended interventions | Mising outcome data | Measurement of the outcome | Selection of the reported result | Overall bias |
| Moore 2021 | low | low | low | low | low | low |

Note: domain1-7 in heading signified: Domain1: Bias due to confounding; Domain2: Bias in the selection of participants; Domain3: Bias in the classification of interventions; Domain4: Bias due to deviations from the intended interventions; Domain 5: Bias due to missing data; Domain 6: Bias in the measurement of outcomes; Domain 7: Bias in the selection of reported outcomes

pooled mPFS and ORR for all included studies were 6.7 months (95% CI: 4.54 to 8.86) and 36% (95% CI: 28–44), respectively. As most of the study cohorts consisted of over treated patients with platinum resistance, the above results can be seen as hopeful.

Due to the significant heterogeneity of the studies included, we believed that performing subgroup analysis on studies with similar designs could provide us with more accurate results. We divided the included studies into two subgroups based on their therapy regimen: MIRV monotherapy and MIRV+BEV combination therapy. There was no heterogeneity between the two subgroups, and the mPFS of the combination therapy group was significantly longer than that of the MIRV monotherapy group (7.78 vs 4.28 months, P = 0.000). Additionally, the pooled ORR for the two subgroups was 43% vs 25% (P = 0.000). These results are in consonance with other studies [30, 32]. Based on these findings, we can tentatively conclude that MIRV and BEV combination therapy is likely a promising antitumour drug combination for solid tumors. GLORIOSA (NCT05445778), a phase III trial testing MIRV and BEV, is currently in the recruiting stage and will provide more specific results about this combination. Moore 2018 [29] used MIRV+CARBO for intervention, and the reported effect indicators were mPFS (15 months) and ORR (71%), which were significantly better than those of MIRV monotherapy or in combination with BEV. Despite the small sample size and the fact that the intervention targeted platinum-sensitive patients, it is still too early to draw conclusions, and more high-quality Randomized Controlled Trials results are needed. Bakes (2021) [23] conducted a study combining MIRV with rucaparib, reporting a favorable mPFS of 6.30 months and an ORR of 38%. Similarly, Moore (2021) [22] reported an mPFS of 4.10 months and an ORR of 22% when MIRV was combined with GEM. However, the sample sizes in both studies were limited, precluding a comprehensive assessment of the efficacy of the drug combinations. Nonetheless, the comparable incidence of adverse reactions in these combinations to those of other drug combinations is a positive sign. We did not perform statistical analysis on the

association between the efficacy of MIRV and the expression of patient FRα receptors primarily because not all included studies provided relevant data. However, according to studies that reported the relevant results, patients with high expression of FRα receptors had a relatively higher response to MIRV[22, 25, 30]. Therefore, in future research, it will be important to identify patients who are more responsive to MIRV, that is, to predict the sensitivity of tumors to MIRV. Some studies have reported initial research findings in this area [33–35].

Platinum-based chemotherapy is a basic drug in tumor treatment; however, platinum resistance is a frequent cause of recurrence in many tumors. The sensitivity of platinum drugs to chemotherapy can be used as a reference for the efficacy of targeted drugs. Therefore, we propose that subgroup analysis based on tumor response to platinum is of significant clinical importance. Our investigation found that while drug usage patterns varied, the overall ORR was 59% in platinum-sensitive patients and 33% in platinum-resistant patients. This suggests that MIRV treatment may provide more benefits to those with platinum-sensitive tumors. However, further high-quality randomized controlled trials are necessary to provide more precise conclusions. Overcoming platinum resistance is a significant challenge in cancer treatment, and various methods and drugs have been investigated to address it [35–38]. Preliminary results have been less promising, but MIRV has shown unexpected successes in treating platinum-resistant solid tumors [36, 37]. The overall ORR for platinum-resistant patients in our study was 33%, which is close to the pooled ORR of 36%. This result is critical for patients with limited treatment options [36], as MIRV treatment may provide an effective solution. Although MIRV did not improve survival in one particular study [39], it did show promise in shrinking tumors in 24% of participants compared to 10% who received standard chemotherapy. These findings serve as inspiration for further research and provide hope for patients facing limited options.

While adverse reactions at level 3/4 were detected in our study, their incidence was relatively low and even lower than the current standard chemotherapy [38, 40, 41]. These findings suggest that the potential risks and hazards associated with MIRV are manageable. MIRV has an important attribute of reducing cytotoxic drug exposure to the bloodstream, which contributes to the reduced side effects compared to other therapeutic drugs [42]. Eye events have been reported as the most common adverse reaction associated with MIRV in previous studies and ongoing discussions [43–45]. Recent articles have focused on preventing, treating, or minimizing the severity of this adverse reaction by implementing close monitoring, early detection, and tailored treatment regimens [46, 47]. Our ultimate goal is to ensure that patients receive maximum benefit from MIRV treatment while minimizing any negative impact on their quality of life. In conclusion, while adverse reactions are an important consideration when using MIRV to treat tumors, our results suggest that the risks are manageable and the side effects are relatively mild compared to other therapeutic drugs. The ongoing research and discussions on how to optimize treatment regimens and minimize adverse reactions will undoubtedly lead to better outcomes for patients.

Our meta-analysis provides a comprehensive and accurate summary of clinical trial data, which is valuable information for informing clinical practice. The subgroup analyses that we conducted increase the potential to identify specific patient subgroups who may benefit from individual treatments, making this study particularly useful for guiding personalized medicine. Sensitivity analyses were carried out to ensure that the results of our meta-analysis are robust and not based on a minority of studies. However, our analysis has some limitations that must be acknowledged. The studies included in our meta-analysis were primarily single-arm studies, which means that selection, measurement and confounding biases can affect the overall quality of the meta-analysis. Confounding bias was identified as a prevalent risk in our assessment. Furthermore, using outcomes such as ORR, mPFS and adverse effects may not capture all

clinical effects of specific treatments, such as overall survival or quality of life. Publication bias can also skew the results of meta-analyses, which should be considered when interpreting the study results [48]. Therefore, it is crucial to carefully consider study design, data sources, and results when interpreting meta-analysis results. In conclusion, while meta-analysis can provide valuable insights for guiding clinical practice, specific limitations must be taken into account, and rigorous methods should be followed to minimize bias and ensure that the results are generalizable.

## 5. Conclusion

This meta-analysis concludes that combining MIRV and BEV significantly improves tumor treatment compared to MIRV alone. Moreover, MIRV shows positive results for platinum-resistant patients, highlighting it as a beneficial treatment option. Adverse reactions to MIRV treatment, such as eye events, are rare and manageable with early monitoring and intervention. Overall, MIRV is an effective and safe treatment option for solid tumors, especially when combined with BEV. This research provides important insights that may be helpful for doctors and healthcare professionals when considering treatment options for patients with solid tumors.

## Supporting information

**S1 Checklist. PRISMA checklist.**
(DOCX)

**S1 Fig. One-left out analysis.**
(TIF)

**S2 Fig. Pooled mPFS after kicked out Moore 2018.**
(TIF)

**S3 Fig. Pooled ORR after kicked out Moore 2018.**
(TIF)

**S1 File. Study protocol.**
(DOC)

**S2 File. All search history.**
(DOCX)

**S3 File. Risk of bias assessment of studies.**
(DOCX)

**S4 File. All data extracted from the primary research sources for the systematic review and meta-analysis.**
(XLSX)

**S5 File. All studies identified in the literature search.**
(XLSX)

**S6 File. An explanation of how missing data were handled.**
(DOCX)

**S7 File. Meta-analysis datasets.**
(DOCX)

## Author Contributions

**Conceptualization:** Shamsnur Rehim, Hongjing Wang.

**Data curation:** Shamsnur Rehim.

**Formal analysis:** Shamsnur Rehim, Shuang Yuan.

**Investigation:** Shamsnur Rehim, Shuang Yuan.

**Methodology:** Shuang Yuan, Hongjing Wang.

**Project administration:** Shamsnur Rehim.

**Software:** Shamsnur Rehim.

**Supervision:** Shuang Yuan, Hongjing Wang.

**Validation:** Hongjing Wang.

**Writing – original draft:** Shamsnur Rehim.

**Writing – review & editing:** Shamsnur Rehim, Shuang Yuan, Hongjing Wang.

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
