## [Decision Letter · Decision Letter 0]

20 May 2024

PONE-D-24-04628Mirvetuximab Soravtansine in Solid Tumors: A Systematic Review and Meta-AnalysisPLOS ONE

Dear Dr. HongJing Wang,

Thank you for submitting your manuscript to PLOS ONE. After careful consideration, we feel that it has merit but does not fully meet PLOS ONE’s publication criteria as it currently stands. Therefore, we invite you to submit a revised version of the manuscript that addresses the points raised during the review process.

We look forward to receiving your revised manuscript.

Kind regards,

Elda Tagliabue

Academic Editor

PLOS ONE

Journal Requirements:

Additional Editor Comments (if provided):

In this study, Shamsnur Rehim al. explored Mirvetuximab Soravtansine efficacy and safety in different cancers through a meta-analysis of different clinical studies. Their findings revealed that this ADC has significant therapeutic effects, especially when combined with Bevacizumab and in cancers expressing high levels of Folate Receptor ɑ. Although this study is of particular interest to the scientific community, the manuscript should be amended according to the reviewer’s concerns.

In addition to spell out words in full at first mention as requested by the reviewer (i.e., PRISMA, HR, RECIST, CTCAE, AE) FRɑ and rucaparib should be specified.

Reviewers' comments:

Reviewer's Responses to Questions

**Comments to the Author**

1. Is the manuscript technically sound, and do the data support the conclusions?

Reviewer #1: Yes

2. Has the statistical analysis been performed appropriately and rigorously? 

Reviewer #1: Yes

3. Have the authors made all data underlying the findings in their manuscript fully available?

Reviewer #1: Yes

4. Is the manuscript presented in an intelligible fashion and written in standard English?

Reviewer #1: Yes

5. Review Comments to the Author

Reviewer #1: The systematic review and meta-analysis by Rehim et al, provide a much needed analysis of the therapeutic benefits of MIRV based on several important variables including prior treatments, platinum response status, monotherapy and combination therapy, and FR-alpha expression levels, as well as adverse drug effects. This type of review is essential for identifying how all these variables affect therapeutic efficacy and safety and what research direction is needed to address outstanding therapeutic concerns. A major limitation of the current study is that many of the studies are not randomized controlled trials and the authors have duly acknowledged this limitation which is not of their own making. Nonetheless, the findings are very interesting. However, I do have several suggestions that will help improve the quality of the manuscript:

1. The data on prior lines of treatment is quite interesting but it would be helpful to further stratify the prior line of treatments based on the actual treatments the patients where receiving, their platinum response status, and their FR-alpha expression levels. This will help eliminate any potential confounding factors preventing us from seeing the importance of prior lines of treatment to the therapeutic effects of MIRV.

2. For table 1, the age should be provided as median IQR, and for the number of prior systemic treatments, what does the n mean? You provide a number and then another number in brackets. That is unclear.

3. It is important to spell out words in full at first mention and then continue using that abbreviation onwards. This is not addressed for PRISMA, HR, RECIST, CTCAE, AE.

4. There is definitely something wrong with the referencing method used in this paper, it is very inconsistent and unclear. The authors are advised to address this issue.

6. PLOS authors have the option to publish the peer review history of their article (what does this mean?). If published, this will include your full peer review and any attached files.

Reviewer #1: No

---

## [Author Response · Author response to Decision Letter 0]

28 May 2024

Response to Reviewer

Dear editor,

Thank you for the opportunity to revise our manuscript, “Mirvetuximab Soravtansine in Solid Tumors: A Systematic Review and Meta-Analysis”, which we submitted to PLOS ONE. We appreciate the feedback from the reviewers and have carefully considered their comments. Below, we provide a point-by-point response to the reviewers' comments and have highlighted the changes made in the revised manuscript.

Reviewer #1: The systematic review and meta-analysis by Rehim et al, provide a much needed analysis of the therapeutic benefits of MIRV based on several important variables including prior treatments, platinum response status, monotherapy and combination therapy, and FR-alpha expression levels, as well as adverse drug effects. This type of review is essential for identifying how all these variables affect therapeutic efficacy and safety and what research direction is needed to address outstanding therapeutic concerns. A major limitation of the current study is that many of the studies are not randomized controlled trials and the authors have duly acknowledged this limitation which is not of their own making. Nonetheless, the findings are very interesting. However, I do have several suggestions that will help improve the quality of the manuscript:

Comment #1: The data on prior lines of treatment is quite interesting but it would be helpful to further stratify the prior line of treatments based on the actual treatments the patients where receiving, their platinum response status, and their FR-alpha expression levels. This will help eliminate any potential confounding factors preventing us from seeing the importance of prior lines of treatment to the therapeutic effects of MIRV.

Response: We agree with the reviewer that stratifying the data based on specific treatments, platinum response status, and FR-alpha expression levels would provide a more nuanced understanding of the impact of prior lines of treatment on the therapeutic effects of Mirvetuximab Soravtansine (MIRV). Following your suggestion, I have further stratified the patients’ actual systemic chemotherapy treatments, drug types exposed during treatment, platinum response status, and FR-α expression levels in the table of main characteristics of the included studies[Table 1]. Unfortunately, only a very small number of studies provided effect estimates based on these subgroups, and no study provided survival data for each patient. Therefore, in our previous statistical analysis, we only conducted subgroup analyses using the data that were available, including patients’ platinum responsiveness, the number of previous systemic chemotherapies, and FR-α expression levels. Indeed, through these subgroup analyses, we were able to observe that these factors do have some impact on the clinical benefits of the drug. The specific analyses are presented in Figures [Figure4,5].

Comment #2: For table 1, the age should be provided as median IQR, and for the number of prior systemic treatments, what does the n mean? You provide a number and then another number in brackets. That is unclear.

Response:We apologize for the lack of clarity in Table 1. We agree that presenting the age as median with the interquartile range (IQR) would be more informative. However, it is regrettable that all the original studies only provided the median age and range, and we were unable to obtain individual patient age data for statistical processing. Regarding the “No. of prior systemic treatments,” the three numbers provided represent the number of prior systemic chemotherapies the patients received, the number of patients who received that number of treatments, and the percentage of these patients within the total study population, respectively. We have updated the table legend to include this explanation, which we hope will provide readers with a clearer understanding.

Comment #3: It is important to spell out words in full at first mention and then continue using that abbreviation onwards. This is not addressed for PRISMA, HR, RECIST, CTCAE, AE.

Response: We acknowledge this oversight and have now ensured that all terms are spelled out in full at their first mention in the manuscript, followed by the corresponding abbreviation, as per the journal’s guidelines. We have corrected this throughout the text to maintain consistency and clarity.

Comment #4: There is definitely something wrong with the referencing method used in this paper, it is very inconsistent and unclear. The authors are advised to address this issue.

Response：We agree that the referencing style in the initial submission was inconsistent and have now standardized the referencing throughout the manuscript to follow the PLOS ONE citation guidelines. We have checked each reference for accuracy and formatting consistency, and we are confident that the revised manuscript adheres to the journal’s standards.

In addition to addressing the specific comments from the reviewers, we have made several other improvements to the manuscript. In Table 1, we have now included the chemotherapy agents to which patients were previously exposed and the FR-α expression levels of the patients in each study. This additional information provides a more comprehensive overview of the patient population and their treatment history, which we believe will enhance the understanding of the study outcomes.

We hope that the revisions we have made address the concerns raised by the reviewers and improve the quality of our manuscript. We are confident that our work is of significant interest to the readers of Plos one and we thank you for considering our work for publication.Please do not hesitate to contact us if you require any further information.

Sincerely,

Dr. Hongjing Wang

Organization: Key Laboratory of Birth Defects and Related Diseases of Women and Children of the Ministry of Education, West China Second University Hospital, Sichuan University, Chengdu, China 

Email address: whjscdx@163.com

---

## [Decision Letter · Decision Letter 1]

6 Sep 2024

Mirvetuximab Soravtansine in Solid Tumors: A Systematic Review and Meta-Analysis

PONE-D-24-04628R1

Dear Dr. HongJing Wang,

We’re pleased to inform you that your manuscript has been judged scientifically suitable for publication and will be formally accepted for publication once it meets all outstanding technical requirements.

Kind regards,

Cho-Hao Howard Lee, M.D.

Academic Editor

PLOS ONE

Additional Editor Comments (optional):

Reviewers' comments:

Reviewer's Responses to Questions

**Comments to the Author**

1. If the authors have adequately addressed your comments raised in a previous round of review and you feel that this manuscript is now acceptable for publication, you may indicate that here to bypass the “Comments to the Author” section, enter your conflict of interest statement in the “Confidential to Editor” section, and submit your "Accept" recommendation.

Reviewer #1: All comments have been addressed

Reviewer #2: (No Response)

Reviewer #3: All comments have been addressed

Reviewer #4: All comments have been addressed

2. Is the manuscript technically sound, and do the data support the conclusions?

Reviewer #1: (No Response)

Reviewer #2: Yes

Reviewer #3: Yes

Reviewer #4: Yes

3. Has the statistical analysis been performed appropriately and rigorously? 

Reviewer #1: (No Response)

Reviewer #2: Yes

Reviewer #3: Yes

Reviewer #4: Yes

4. Have the authors made all data underlying the findings in their manuscript fully available?

Reviewer #1: (No Response)

Reviewer #2: Yes

Reviewer #3: Yes

Reviewer #4: Yes

5. Is the manuscript presented in an intelligible fashion and written in standard English?

Reviewer #1: (No Response)

Reviewer #2: Yes

Reviewer #3: Yes

Reviewer #4: Yes

6. Review Comments to the Author

Reviewer #1: (No Response)

Reviewer #2: Strengths:

Important Topic: I personally feel this study focuses on a very important and timely subject. Mirvetuximab Soravtansine (MIRV) is a novel treatment that has garnered attention recently, especially after its FDA approval. So, it's quite valuable to gather more comprehensive data about its efficacy and safety. The authors made a great choice in doing this systematic review and meta-analysis.

Extensive Literature Search: It’s really impressive how the authors combed through several reputable databases like PubMed and Cochrane. They didn’t leave any stones unturned when looking for studies. I think this strengthens the paper's reliability, especially since it includes a broad range of solid tumors.

Methodology is Sound: The authors followed the PRISMA guidelines, which I think adds to the strength of this work. They applied a random-effects model, which was quite appropriate, considering the heterogeneity. It shows that the team is aware of the technical demands of a meta-analysis.

Well-presented Results: The results were presented quite clearly. I especially liked the figures—particularly the forest plots—they made it easier to digest the data. It was well thought out, and I believe the readers will appreciate that too.

Clinical Relevance: The focus on the combination therapy of MIRV with Bevacizumab is quite useful, especially since platinum-resistant tumors are so hard to treat. The authors’ insights here seem practical for guiding future treatments.

Suggestions for Improvement:

High Heterogeneity: One thing I noticed right away is that the study suffers from quite high heterogeneity (I² values are up to 96%). It’s good that the authors used a random-effects model, but I personally feel they should dive a little deeper into why this heterogeneity exists. Maybe, they can discuss if doing more sensitivity analyses or meta-regressions could help.

Table Clarifications: Table 1 is quite confusing. The number of prior treatments needs better explanation, and even though the authors tried to explain in their response, it’s still a bit unclear to me. Maybe add more details in the legend to ensure readers get it right on the first glance.

Referencing Style Needs Attention: I think the referencing system could use some polish. It’s a little inconsistent throughout the manuscript, and I noticed that some citations don’t quite fit the style used elsewhere. The authors should ensure consistency for a better flow.

FRα Subgroup Analysis: The subgroup analysis on FRα expression is insightful, but honestly, there’s not enough data to support solid conclusions here. I’d recommend they mention the limitations more explicitly, so future researchers understand the gaps that still need to be filled.

Adverse Events: The discussion around adverse effects is good, but it could be better if they expanded a bit more on management strategies for common side effects, especially ocular toxicity. Given how prevalent it seems to be with MIRV, that would be a helpful addition.

Risk of Bias: I appreciate that the authors acknowledge the majority of studies included are single-arm trials, which obviously increases bias risk. But I feel they could discuss the influence of this bias in more detail. It would make the paper more robust in acknowledging its limitations.

Conclusion:

All in all, this is a solid paper that covers an important topic. I would personally recommend minor revisions, particularly around clarifying the tables and improving the discussion on heterogeneity and bias. Once these are addressed, I think it will be a valuable addition to the literature. It’s clear the authors have put in a lot of effort and attention to detail, so just a few tweaks are needed before this can be published.

Reviewer #3: The manuscript "Mirvetuximab Soravtansine in Solid Tumors: A Systematic Review and Meta-Analysis" is a high-quality work that provides valuable insights into the therapeutic potential of MIRV in solid tumors. I appreciate that the manuscript thoroughly addresses all previous comments. The authors have also ensured consistency in terminology and reference formatting, which meets the journal’s standards.

Given the thoroughness of the analysis and the overall quality of the manuscript, I see no need for further revisions. I commend the authors for their meticulous work and believe this manuscript makes a significant contribution to the literature on targeted therapies for solid tumors. I recommend it for publication as it stands.

Reviewer #4: I have reviewed the manuscript and I find it to be a well-executed and comprehensive meta-analysis that addresses an important area of research. The authors have systematically analyzed the efficacy and safety of Mirvetuximab Soravtansine in solid tumors, providing useful insights, particularly regarding the impact of different therapy regimens, platinum sensitivity, and FR-alpha expression levels. The manuscript is generally well-structured, with the results clearly presented and the discussion thoughtfully addressing the strengths and limitations of the study. The authors have made a commendable effort to stratify data and perform subgroup analyses, which adds depth to the findings. The statistical analysis are also sound.

7. PLOS authors have the option to publish the peer review history of their article (what does this mean?). If published, this will include your full peer review and any attached files.

Reviewer #1: No

Reviewer #2: **Yes: **Xiaoyi Zhang, MD

Reviewer #3: **Yes: **Yizhe Song

Reviewer #4: **Yes: **Yuhang Liu

---

## [Editor Report · Acceptance letter]

21 Sep 2024

PONE-D-24-04628R1 

PLOS ONE

Dear Dr. Wang, 

I'm pleased to inform you that your manuscript has been deemed suitable for publication in PLOS ONE. Congratulations! Your manuscript is now being handed over to our production team.

Kind regards, 

on behalf of

Dr. Cho-Hao Howard Lee 

Academic Editor

PLOS ONE